# Blood Biomarkers in Patients with Parkinson’s Disease: A Review in Context of Anesthetic Care

**DOI:** 10.3390/diagnostics13040693

**Published:** 2023-02-12

**Authors:** Jin Joo, Jongmin Jeong, Hue Jung Park

**Affiliations:** Department of Anesthesiology and Pain Medicine, Seoul St. Mary’s Hospital, College of Medicine, The Catholic University of Korea, Seoul 06591, Republic of Korea

**Keywords:** Parkinson’s disease, inflammatory biomarkers, surgery, anesthesia, blood-brain barrier

## Abstract

Parkinson’s disease (PD) is the second most common inflammatory neurodegenerative disorder after dementia. Preclinical and epidemiological data strongly suggest that chronic neuroinflammation slowly induces neuronal dysfunction. Activated microglia secrete several neurotoxic substances, such as chemokines and proinflammatory cytokines, which may promote blood–brain barrier (BBB) permeabilization. CD4^+^ T cells comprise proinflammatory cells such as T helper (Th) 1 and Th17 cells, as well as anti-inflammatory cells such as Th2 and T regulatory cells (Tregs). Th1 and Th17 cells can be detrimental to dopamine neurons, whereas Th2 and Tregs are neuroprotective. The results of studies on the serum levels of cytokines such as IFN-γ and TNF-α secreted by Th1 T cells, IL-8 and IL-10 secreted by Th2 T cells, and IL-17 secreted by Th17 cells in PD patients are not uniform. In addition, the relationships between serum cytokine levels and motor and non-motor symptoms of PD are controversial. Surgical stress and anesthesia induce inflammatory responses by disturbing the balance between pro- and anti-inflammatory cytokines, which may exacerbate the neuroinflammatory response in PD patients. Here we review studies on blood inflammatory biomarkers in PD patients and discuss the roles of surgery and anesthesia in PD progression.

## 1. Introduction

Parkinson’s disease (PD) is the second most common inflammatory neurodegenerative disorder after dementia, affecting 7–10 million people worldwide [1]. About 2–3% of elderly patients (aged ≥ 65 years) are affected by PD [2]. People with PD suffer motor symptoms such as bradykinesia, rigidity, resting tremor, and postural instability. In addition, patients often complain of ‘non-motor symptoms’ such as cognitive impairment, anxiety, depression, hypothermia, constipation, bowel and rapid eye movement sleep behavior disorder, and autonomic nervous system disorders [2,3].

PD is characterized by the progressive loss of dopaminergic neurons in the substantia nigra pars compacta, the appearance of Lewy bodies (intracellular inclusions of aggregated a-synuclein), and the presence of neuroinflammation [4,5,6,7]. Although the mechanism of neuron loss in PD is unclear, inflammation and the peripheral immune system play key roles [7,8,9]. Preclinical and epidemiological data suggest that chronic neuroinflammation induces neuronal dysfunction during the asymptomatic stage of PD [6,10]. The activation of resident microglia precedes dopamine neuron loss [11,12]. Activated microglia secrete several neurotoxic substances, such as chemokines and proinflammatory cytokines, which may cause blood–brain barrier (BBB) permeabilization and subsequent infiltration of peripheral leukocytes into the central nervous system (CNS) [11,12,13].

There is currently no cure for PD to prevent PD or delay its progression, mainly due to the still limited comprehension of the events ultimately leading to neurodegeneration. Available treatments for PD are only symptomatic, aimed at relieving the loss of brain dopaminergic neurons by using levodopa (the dopamine precursor), some dopaminergic agonists, and other indirect dopaminergic agents. Surgery, including deep brain stimulation, may be considered in advanced PD patients who fail to respond to levodopa [14].

CD4^+^ T lymphocytes play a pivotal role in orchestrating immune responses implicated not only in the pathogenesis of inflammatory diseases, but also in host defense. CD4^+^ T cells include proinflammatory cells such as T helper (Th) 1 and Th17, and anti-inflammatory cells such as Th2 and T regulatory cells (Tregs) [15,16]. Interestingly, both animal models of PD and clinical studies suggest that Th1 and Th17 can be detrimental to dopamine neurons, whereas Th2 and Tregs are neuroprotective [17,18]. Indeed, the number of circulating CD4^+^ T cells is reduced in patients with PD [19], but the relative proportions and functional profiles of subdivided cell populations are controversial. The decrease in CD4^+^ T cells seen in the peripheral blood of patients with PD is mainly due to decreases in Th2, Th17, and Tregs [20,21]. As a result, Th1 T cells, the absolute numbers of which are similar to healthy controls, increase in PD patients compared to other T cells, resulting in a Th1 bias. Consequently, the production of IFN-γ and TNF-α by Th1 cell lineages increases [21]. The results of studies on the serum levels of cytokines such as IFN-γ and TNF-α secreted by Th1 T cells, IL-8 and IL-10 secreted by Th2 T cells, and IL-17 secreted by Th17 in PD patients are not uniform [19,22,23,24,25,26,27]. In addition, the relationships between serum cytokine levels and motor and non-motor symptoms of PD are controversial [19,23,26,27]. 

Surgical stress and anesthesia induce inflammatory responses by disturbing the balance between pro- and anti-inflammatory cytokines [28], which may exacerbate the neuroinflammatory response in PD patients. The effects of inhalational anesthetics on the inflammatory response are controversial [29,30,31,32]. Meanwhile, Shan et al. [33] reported that sevoflurane worsened the prognosis of PD in a *Drosophila* model. There are few studies on the effect of the immune response on PD symptoms and prognosis after surgery and anesthesia [34,35,36,37]. 

Here we review studies on blood inflammatory biomarkers in PD patients and discuss the role of surgery and anesthesia in PD progression. We searched the PubMed, PubMed Central, Medline, Google Scholar, and Google databases for clinical studies using the key words “Parkinson’s disease”, “inflammation”, “immune” “blood”, “cytokine”, and “chemokine.”

## 2. Blood Inflammatory Biomarkers in PD Patients

Only seven markers (CRP, IL-1β, IL-2, IL-6, IL-8, IFN-γ, TNF-α) were reported in more than 5 of 51 studies. The most frequently studied inflammatory biomarkers are CRP and IL-1β [26,38,39,40,41,42,43,44,45,46,47,48,49,50,51,52,53,54,55,56,57,58]. These two biomarkers show consistently higher levels in PD patients than healthy controls. Data for IL-2, IL-6, IFN-γ, and TNF-α are controversial. Some studies report higher blood levels of these biomarkers, while others found no differences between PD patients and healthy controls, or lower levels in the former group [27,38,41,42,43,44,49,50,51,53,54,55,56,59,60,61,62,63,64,65,66,67,68,69,70,71,72]. Blood markers evaluated to date are listed in Table 1 and Figure 1.

Although many studies have focused on proinflammatory profiles, inflammation is a balance between pro- and anti-inflammatory processes. CD4^+^ T lymphocytes orchestrate an effective immune response during host defense, as well as in the pathogenesis of inflammatory diseases. CD4^+^ T cells can select for proinflammatory phenotypes such as Th 1 and Th17 cells, as well as anti-inflammatory phenotypes such as Th2 and Tregs [15,16]. Results from animal models of PD and clinical studies suggest that Th1 and Th17 cells are detrimental to neurons, while Th2 and Tregs are neuroprotective [17,18].

The most investigated anti-inflammatory blood biomarker in the context of PD is IL-10. However, while some studies report higher levels in PD patients, others report lower levels or no difference in PD patients compared to healthy controls [41,43,44,49,52,53,54,60,67,70].

## 3. Effects of Blood Biomarkers on PD Symptoms

High blood levels of CRP, IL-6 [45,56,62,77,87], fractalkine, IL-1β, IL-15, IFN-γ, S100B, TNF-α, and VCAM-1 [81] are related to worse motor function [58,69,74,78,80,82]. By contrast, reduced blood levels of the anti-inflammatory markers IL-4 and IL-12p40 are associated with worse motor function [44,90]. 

CRP, FABP, IL-6, IL-10, IL-17A, TNF-α, and TNFR are related to worse cognitive function and dementia [68,77,80,83]. Furthermore, higher levels of C3 and C4 are related to reduced memory function [23,88]. Dufek et al. [75] showed an association between a high level of IL-6 and overall mortality in PD.

## 4. Influence of Peripheral Inflammation on the CNS

Systemic inflammation causes physiological and behavioral changes in humans and animals, characterized by reduced cognitive function, fever, reduced food intake, drowsiness, and general fatigue [91]. Aseptic inflammation activates innate immune pathways similar to other forms of immunologic attack [92]. Importantly, although inflammation is protective against injury overall, it can be detrimental if dysregulated and can contribute to pathologies including neuroinflammation [93,94,95]. The impact of systemic inflammation on the brain can be severe. Blood-borne factors, as well as a systemic proinflammatory milieu, degrade CNS function, thereby directly affecting synaptic plasticity and cognitive function during normal aging [96,97].

Endothelial cells, pericytes, and astrocyte end-feet are key components of the neurovascular unit [98]. Together with tight junction and adherent proteins of the endothelial cell layer, they ensure proper barrier formation and protection against potentially harmful peripheral molecules [99]. Under pathological conditions, the BBB allows extravasation of various immune cells and systemic biomarkers, such as plasma proteins, prostaglandins, cytokines, and chemokines, into brain parenchyma [100]. Surgery triggers inflammation and receptors expressed on the BBB can lead to endothelial inflammation and subsequent neuroinflammation [101,102,103,104,105]. 

Cytokines and the migration of peripheral immunocompetent cells across the BBB are associated with perioperative neurocognitive impairment in animal models [105]. Other preclinical surgical models have shown similar changes in the BBB ultrastructure, with exogenous tracers penetrating into the brain parenchyma [102,106]. In aged mice, laparotomy may trigger changes in several markers including claudins, occludins, and adhesion molecules, leading to an increase in BBB permeability. The increase in BBB permeability resulted in cognitive decline in a manner dependent on IL-6 signaling [104]. Furthermore, the administration of an IL-6 monoclonal antibody and targeting of TNF-α prevent perioperative neurocognitive disorder [103,107]. Surgery can upregulate enzymes that degrade extracellular matrix 9, such as matrix metallopeptidases, and cause the increase in BBB permeability and neuroinflammation [101]. Different concentrations of sevoflurane anesthesia differentially regulate matrix metallopeptidase 9 and 2 [108], suggesting that anesthesia itself contributes to the increase in BBB permeability. The role and timing of BBB/neurovascular unit opening after surgery need further investigation to develop strategies to suppress neuroinflammation during surgery.

## 5. Impact of Surgery and Anesthesia on the Immune System

Tissue damage during major surgery induces inflammatory responses by disturbing the balance between pro- and anti-inflammatory cytokines [28]. Other invasive measures such as mechanical ventilation, transfusion of blood products, and perioperative tissue hypoperfusion with reperfusion injury can also trigger a perioperative immune response [92]. 

Neutrophils are the primary effector cells of the innate immune system and represent the first line of defense against invading exogenous pathogens [109]. Neutrophil functions also carry a risk of host tissue damage, which may lead to organ dysfunction, making neutrophil activation a double-edged sword [110]. Neutrophils are also recruited and activated during aseptic inflammation, as seen in cases of ischemia-reperfusion injury and excessive tissue damage caused by surgical procedures. After surgical procedures, oxidative stress is evoked by activated neutrophils in association with inflammatory processes [111]. Aseptic inflammation activates the innate immune response by releasing cytokines [92]. Blood levels of IL-1ß, IL-6 and IL-10 are commonly elevated after tissue damage [112,113,114]. IL-6 (a proinflammatory factor) and IL-10 (an anti-inflammatory factor) are simultaneously increased, indicating the activation of opposite mechanisms. Although IL-6 is proinflammatory, it can also inhibit Th1 differentiation [115]. It is therefore conceivable that IL-6 also suppresses host defenses.

Most general anesthetic agents directly or indirectly suppress the immune response [116,117,118], including by impairing neutrophil function [119]. Modulation of neuro-immunomodulatory circuits is another immunosuppressive mechanism of anesthetics [120]. Total intravenous anesthesia using propofol is superior to inhalational anesthetics for inhibiting inflammatory responses [121,122,123,124]. The effects of inhalational anesthetics on the inflammatory response are controversial [29,30,31,32]. Regional anesthesia is used to partially alleviate the stress response to the surgical procedure. A retrospective study showed fewer perioperative complications with regional compared to general anesthesia [125]. Avoidance of general anesthesia can reduce the perioperative immune response and serum cortisol level [126]. Liu et al. [127] reported that single use of regional anesthesia was associated with a lower postoperative infection rate compared to general anesthesia. Elevation of inflammatory biomarkers has been noted after general and spinal anesthesia [128]. The effects of different methods of anesthesia on the immune response have been documented in studies on cancer recurrence and metastasis [129,130,131]. However, the effects of different methods of anesthesia on clinical outcomes are unclear.

## 6. Influence of Surgery and Anesthesia on the Prognosis of Parkinson’s Disease

There has been diverse research reporting that dysbiosis, obesity, diabetes, and vascular disorders are closely related to the occurrence of neurodegenerative diseases including Parkinson’s disease [132,133,134,135,136]. However, few studies have reported worsening symptoms after surgery and anesthesia in PD patients [34,36,37]. After total knee arthroplasty, functional ability does not improve in patients with PD, and functional loss has been reported in final-stage PD patients. Poorer knee function has been found in PD patients compared with healthy controls [36].

Shan et al. [33] reported that impairment of locomotor abilities was exacerbated by exposure to sevoflurane in a *Drosophila* PD model. Tuon et al. [137] reported that IL-17 levels decreased after physical training in an experimental mouse model of PD. Serum levels of cytokines, such as IL-1β, TNF-α, and IL-10, were higher in PD patients than age-matched non-PD controls; higher TNF-α levels were associated with a faster rate of motor decline and higher IL-1β levels with a faster rate of cognitive decline [22]. Therefore, IL-17 and TNF-α are closely related to the motor symptoms of PD. In a study of patients diagnosed with PD about 10 years prior, deep brain stimulation surgery was required to manage motor symptoms that were not controlled by medications. These observations suggest that serum levels of IL-17 increase with the progression of PD. The larger increase of IL-17 at 24 h after surgery seen under inhalation anesthetics suggests that propofol has advantages in terms of the inhibition of neuroinflammation after surgery [138]. However, studies on the effects of surgery and anesthesia on neuroinflammation and consequent worsening of PD symptoms are scarce. As the elderly population increases, the number of patients with PD who undergo surgery is also growing. Therefore, studies should evaluate neuroinflammatory changes and symptom exacerbation in PD patients during the perioperative period.

## 7. Conclusions

Insight into the inflammatory processes induced by surgical and therapeutic interventions during the perioperative period has been provided by clinical and experimental studies. Because inflammation affects disease progression and patient outcomes at multiple levels, knowledge of the molecular mechanisms and pathways involved in these processes would facilitate the development of new therapeutic approaches. CRP, IL-1β, IL-2, IL-6, IL-8, IFN-γ, TNF-α are 7 the most intensely evaluated blood biomarkers in PD patients. Although they are correlated with motor and nonmotor symptoms, the results are controversial and vary by study design. Surgery and anesthesia induce inflammatory responses by disturbing the balance between pro- and anti-inflammatory cytokines. Studies on the effects of surgery and anesthesia on neuroinflammation and the consequent worsening of PD symptoms are scarce. Therefore, research on the effects of surgery and anesthesia on neuroinflammation, and methods for its amelioration, are needed to prevent worsening of perioperative symptoms in patients with PD.

## Figures and Tables

**Figure 1 diagnostics-13-00693-f001:**
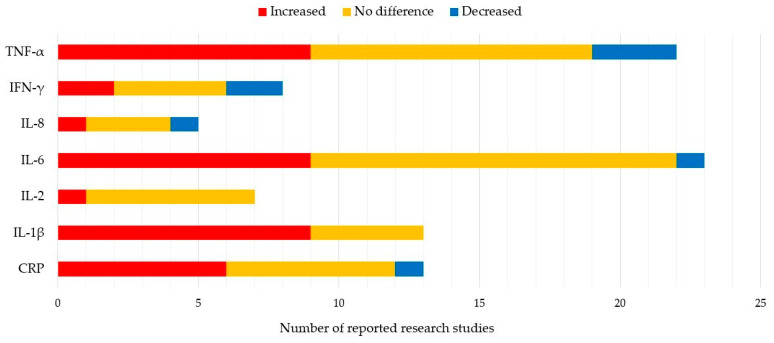
7 Most reported blood biomarkers and their value relative to healthy controls.

**Table 1 diagnostics-13-00693-t001:** Blood biomarkers and their related symptoms.

Clinical Trials	Blood Markers	Related Symptoms
Alrafiah et al. [55]	IL-1β, IL-6, TNF-α	
Andican et al. [39]	CRP, ICAM-1	
Bagheri et al. [73]	CXCL 12, CXCR4	
Baran et al. [46]	CRP, HMGB1	
Brockmann et al. [52]	FABP, IL-10, IL-12p40, SCF, BDNF	
Calvani et al. [65]	MIP-1α, MIP-1β, IL-8, IL-9	
Carvalho et al. [74]	S100B *	nonmotor
Chatterjee et al. [57]	IL-1β, NLRP3	
Csencsits-Smith et al. [71]	MCP-1, IP-10, TNF-α	
Delgado-Alvarado et al. [62]	IL-6 *	motor, nonmotor
Dommershuijsen et al. [48]	CRP	
Dufek et al. [75]	IL-6 *	mortality
Dumitrescu et al. [76]	calprotectin	
Eidson et al. [69]	IL-8 *, IFN-γ, NGAL, TNF-α	motor, nonmotor
Fan et al. [58]	IL-1β *, NLRP3	motor, nonmotor
Green et al. [77]	IL-6 *, IL-17A *, TNF-α, TGF-β	motor, nonmotor
Gupta et al. [78]	Fractakine *, 3-NT *	motor
Gupta et al. [27]	IL-8, TNF- α	
Herlofson et al. [79]	IL1-Ra * , VCAM-1 *	nonmotor
Hu et al. [50]	IL-1β, TNF-α	
Jin et al. [47]	CRP	
Karpenko et al. [53]	IL-1β, IL-1Ra, IL-6, IL-10 *, TNF-α *	nonmotor
Kim et al. [43]	CRP, IL-1β, IL-2, IL-6, IL-10 *, TNF-α	nonmotor
King et al. [44]	CRP, IL-2 *, IL-4, IL-6, IL-8, IL-10, IFN-γ, TNF-α	motor
Kouchaki et al. [72]	IL-27 *, TNF-α *	motor
Koziorowski et al. [49]	IL-1β, IL-6 *, IL-10, IL-12, TNF-α, NT-proCNP	motor
Kwiatek-Majkusiak et al. [63]	IL-6	
Lerche et al. [80]	FABP *, TNF-α *, CA-125 *, BDNF*	motor, nonmotor
Lian et al. [56]	IL-1β, IL-6 *	motor
Lin et al. [66]	IL-1β, IL-2, IL-4, IL-6, IL-13, IL-18, IFN-γ, TNF-α	
Lindqvist et al. [26]	CRP, IL-6 *, sIL-2R, TNF-a	nonmotor
Mahlknecht et al. [59]	MCP-4, ICAM-1, IL-2, IL-6, Leptin, PDGF-BB, prolactin, RANTES	
Martin-Ruiz et al. [70]	CRP *, IL-6 *, IL-10, TNF-α	nonmotor
Miliukhina et al. [64]	MCP-1, IL-1β, IL-2, IL-4, IL-6, IL-10, IL-12, IL-13, IL-21, IL-23, INF-γ, TNF-α	
Milyukhina et al. [51]	IL-1β *, IL-6, IL-10 *, TNF-α *	nonmotor
Pereira et al. [81]	IL-6 *	nonmotor
Perner et al. [82]	VCAM-1 *	motor
Rathnayake et al. [67]	IL-10, IFN-γ, TNF-α	
Rocha et al. [54]	IL-1β, IL-2, IL-4, IL-6, IL-10, IL-17A, IFN-γ, TNF-α	
Rocha et al. [83]	sTNFR1 * , sTNFR2 *	nonmotor
Roy et al. [84]	NLRP3	
Santos-Garcia et al. [45]	CRP *	motor
Sawada et al. [85]	CRP *	nonmotor
Schroder et al. [60]	IL-2, IL-4, IL-5, IL-6, IL-9, IL10, IL-13, IL-17A, IL-17F, IL-21, IL-22, IFN-γ, TNF-α, 1111, CCL17, CCL20, CXCL1, CXCL5, CXCL9, CXCL11, IL-8, IP-10, MCP-1, MIP-1α, MIP-1β, RANTES	
Sun et al. [86]	C3 *, C4	nonmotor
Tang et al. [61]	RANTES *, IL-6 *	motor
Ton et al. [38]	CRP, IL-6	
Umemura et al. [87]	CRP *	motor
Usenko et al. [68]	MCP-1, IL-4, IL-6, IL-10*, INF-γ, TNF-α *	nonmotor
Vesely et al. [23]	C3 *, C4*, IL-6 *	nonmotor
Vesely et al. [88]	C3 *	nonmotor
Wang et al. [42]	CRP *, IL-1β, sIL-2R *, IL-6, IFN-γ, TNF-α*	nonmotor
Yang et al. [89]	IL-6, IL-10, IL-17, IL-23, TGF-β	
Yilmaz et al. [90]	IL-12p40 *	motor, nonmotor

BDNF, brain-derived neurotrophic factor; C, complement; CA-125, cancer antigen 125; CCL, C-C motif chemokine; CRP, C-reactive protein; CXCL 12, C-X-C motif chemokine ligand 12; CXCR4, C-X-C chemokine receptor type 4; FABP, fatty acid-binding protein; HMGB1, high-mobility group box 1 protein; ICAM-1, intercellular adhesion molecule 1; IL1-Ra, interleukin-1 receptor antagonist; IP-10, interferon gamma-induced protein 10; IFN-γ, interferon gamma; MCP, monocyte chemoattractant protein-1; 3-NT, 3-nitrotyrosine; MIP, macrophage inflammatory protein; NGAL, neutrophil gelatinase associated lipocalin; NLRP3, NLR family pyrin domain containing 3; NT-proCNP, N-terminal pro c-type natriuretic peptide; PDGF-BB, platelet-derived growth factor-BB; RANTES, regulated upon activation, normal T-cell expressed and presumably secreted; S100B, S100 calcium binding protein B; SCF, stem cell factor; sIL-2R, soluble interleukin-2 receptors; sTNFR, soluble tumor necrosis factor receptors; TGF-β, transforming growth factor-beta; TNF-α, tumor necrosis factor alpha; VCAM-1, vascular cell adhesion protein 1; red colored, increased in Parkinson’s disease patients relative to healthy controls; blue colored, decreased in Parkinson’s disease patients relative to healthy controls; green colored, positively correlated with symptom; orange colored, negatively correlated with the symptom; *, significantly correlated with the symptom.

## Data Availability

Not applicable.

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
