# Peer review of "Blood Biomarkers in Patients with Parkinson’s Disease: A Review in Context of Anesthetic Care"

_diagnostics, 2023, doi:10.3390/diagnostics13040693_

Round 1

Reviewer 1 Report

Table 1 is very exhaustive and demonstrates the diligence of the author in identifying the studies and the presumed roles of the cytokines. However, it is quite difficult to follow, and does not contribute significantly to the general information, as it is impossible to remember which marker is increased or decreased and what motor effects it has.

A suggestion would be to replace it with a synthetic figure where the markers could be color-coded and the increased or decreased values could be assigned numeric values, making it more suggestive…

Line 136 " BBB opening and cognitive decline" do not go well in the same phrase, as the first is a pathophysiological occurence and the second is a determination of function. Also, the notion of "BBB opening" should be previously defined as a pathophysiological entity, because we do not know exactly what it entails (pages 140, 142,143). (Maybe one line.)

The paper is very well-made, however, due to the abundance of acronyms and various makers, it is quite difficult to read. A few graphs might enhance the retention after reading, mainly before the conclusions.

Author Response

Point 1: Table 1 is very exhaustive and demonstrates the diligence of the author in identifying the studies and the presumed roles of the cytokines. However, it is quite difficult to follow, and does not contribute significantly to the general information, as it is impossible to remember which marker is increased or decreased and what motor effects it has.

A suggestion would be to replace it with a synthetic figure where the markers could be color-coded and the increased or decreased values could be assigned numeric values, making it more suggestive…

Response 1: As you pointed out, I admit that Table 1 is complex and somewhat hard to understand. So, we clored the cytokines according to their values; increased cytokines in red, decreased cytokines in blue, cytokines with a positive relationship to symptoms in green, and cytokines with a negative relationship in orange. In addition, the results of the 7 most studied cytokines were separately graphed and set as Figure 1.

Point 2: Line 136 " BBB opening and cognitive decline" do not go well in the same phrase, as the first is a pathophysiological occurence and the second is a determination of function. Also, the notion of "BBB opening" should be previously defined as a pathophysiological entity, because we do not know exactly what it entails (pages 140, 142,143). (Maybe one line.)

Response 2: In order to clarify the meaning of BBB opening, it was replaced with the phrase ‘increased BBB permeability’. In addition, a new sentence related to cognitive decline was created to make the meaning of the sentence concise and clear.

Reviewer 2 Report

a few comments about device-aided therapies in advanced PD

Author Response

Point 1: a few comments about device-aided therapies in advanced PD

Response 1: As you recommended, we have added sentences related to the treatment of PD including surgical management in the Introduction.

Reviewer 3 Report

The manuscript is well structured and summarizes recent literature data. Just a note, anesthesia and surgery can contribute to systemic inflammation but represent a normal process of repair and healing, they represent a problem in subjects with compromised clinical conditions, such as the elderly, diabetics, heart patients or immunosuppressed. In the latter case they tend to exacerbate the already present low-grade inflammation, but there is no evidence that they could contribute to neurodegenerative diseases, with the exception of those listed for HIV-induced immunocompromise or not

Accordingly, I would remove these paragraphs, and add other conditions that are well known to contribute to neurogenerative diseases such as Parkinson's, including dysbiosis, endotoxemia, obesity and diabetes, or vascular disorders

Author Response

Point 1: The manuscript is well structured and summarizes recent literature data. Just a note, anesthesia and surgery can contribute to systemic inflammation but represent a normal process of repair and healing, they represent a problem in subjects with compromised clinical conditions, such as the elderly, diabetics, heart patients or immunosuppressed. In the latter case they tend to exacerbate the already present low-grade inflammation, but there is no evidence that they could contribute to neurodegenerative diseases, with the exception of those listed for HIV-induced immunocompromise or not

Accordingly, I would remove these paragraphs, and add other conditions that are well known to contribute to neurogenerative diseases such as Parkinson's, including dysbiosis, endotoxemia, obesity and diabetes, or vascular disorders

Response 1: As you said, there are factors that have been shown to contribute to neurodegenerative diseases including PD. In fact, the effects of surgery and anesthesia on neuroinflammation are not yet well understood. However, as I mentioned in Section 5, recent studies have been actively published showing that immune changes caused by surgery and anesthesia affect cancer recurrence and metastasis. In this review paper, I tried to summarize the research reports that immune changes caused by surgery and anesthesia affect neuroinflammation, and talk about the need for future research on the effects of these changes on patients with PD. So, as you pointed out, I mentioned the factors affecting neurodegenerative diseases with references and added in Section 6 and that there are few studies on anesthesia and surgery yet.